# Fruit quality and antioxidant potential of *Prunus humilis* Bunge accessions

**Hongbo Fu**[1,2,3], **Xiaopeng Mu**[1], **Pengfei Wang**[1], **Jiancheng Zhang**[1], **Baochun Fu**[3], **Junjie Du**[1] *

**1** College of Horticulture, Shanxi Agricultural University, Taigu, Shanxi, People's Republic of China, **2** Rural Revitalization Institute of Science and Technology, Heilongjiang Academy of Agricultural Sciences, Harbin, Heilongjiang, People's Republic of China, **3** Research Institute of Pomology, Shanxi Agricultural University, Taigu, Shanxi, People's Republic of China

* djj738@163.com

**Data Availability Statement:** All relevant data are in the manuscript and its Supporting Information files.

## Abstract

In this study, we aimed to evaluate the fruit quality of *Prunus humilis* and identify cultivars that could provide superior human health benefits. We measured the basic characteristics, bioactive compounds, and antioxidant capacities of 137 *P. humilis* accessions. Flavonoid and phenol content were determined via colorimetry and ultrahigh performance liquid chromatography. Single fruit and stone weights varied widely and were genetically diverse among accessions. The variation in soluble solid content was comparatively narrow. Total flavonoid content (TFC) ranged from 3.90 to 28.37 mg/g FW, with an average of 10.58 mg/g FW in 2019. Significant differences between accessions in terms of TFC, total phenol content, and antioxidant capacity were found. TFC in the accessions was normally distributed and predominantly in the medium range (9.57–15.23 mg/g FW). Red was the predominant peel color over all other phenotypes (i.e., dark red, red, light red, red-orange, and yellow). There was no obvious correlation between peel color and TFC. Catechin was the major flavonoid component in the fruit. Principal component analysis showed that TFC, ABTS, single fruit weight, and vertical and horizontal diameter contributed to the first two principal components for each accession. Accessions 10–02, 3-17-2, 3-17-4, and JD1-6-7-37 were characterized by high TFC, ABTS, and large fruit. We believe that our results will aid in the breeding and functional food processing of *Prunus humilis*.

## Introduction

*Prunus humilis* Bunge (*Cerasus humilis* (Bunge) S.Ya.Sokolov) (Rosaceae) is a small deciduous shrub [1]. Like peach, plum, and apricot, it is an ancient tree species in China with a cultivation history dates back to around 3000 years [2]. In the north of China, it is recorded from about 13 provinces, including Shanxi, Hebei, and Liaoning [3]. The root of *P. humilis* is well developed and has strong ecological adaptability [4]. *Prunus humilis* can grow on barren land, withstand drought, and conserve soil water. It has been used by the Forestry Bureau in the Sand Control Project around Beijing and Tianjin areas and in the Three-North Shelter Forest Program [5].

**Funding:** This work was supported by the Key Research and Development Projects of Shanxi Province (Grant No. 201703D221028-4), the Key Projects of Key Research and Development of Shanxi Province (Grant No. 201703D211001-04-04), and the Applied Basic Research Project of Shanxi Province (Grant No. 201801D121251).

**Competing interests:** The authors have declared that no competing interests exist.

The fruits of *P. humilis* are known as 'calcium fruit' in China due to the high calcium content [6]. The fruits are also rich in amino acids, vitamins, organic acids, and mineral elements and contain relatively high levels of anthocyanins, flavanols, flavonols, tannins, and other substances [7]. They can be consumed fresh or used in wine, juice, jam, and several other products. This plant has both ecological and economic value. It is highly nutritious, resistant to various abiotic and biotic stressors, and has human health benefits. It has great research, development, and utilization potential [3].

As human standards of living continue to improve, the importance of fruit quality among both producers and consumers has increased. Fruit quality parameters include color, flavor, shape, size, taste, and texture. These criteria are used to assess fruit commodity value and substantially influence consumer preference [8,9]. The acid and sugar content and proportion affect fruit taste [10,11] and determine fruit quality. Flavonoids are polyphenols and important secondary metabolites in plants. They possess strong biological activity and affect fruit color and flavor [12]. Additionally, flavonoids have antioxidant, antiaging, antiviral, and antitumor properties. They inhibit mutagenesis and enhance microcirculation [13–15]. Free radicals are associated with many human diseases [16]. An increasing incidence of non-communicable diseases, all closely associated with oxidative stress, is motivating scientists to look for natural disease prevention method. Fruits are an important component of traditional food, and are also essential items in a healthy diet for the modern urban population [17]. Moreover, in this view fruits such as *Prunus humilis*, represents potent sources of bioactive compounds, with strong health-promoting and disease-preventing activities [18]. These compounds have a strong antioxidant capacity and also can scavenge free radicals. Foods derived from plant materials with high antioxidant activity are currently receiving much attention and constitute a new trend in plant resource utilization [19].

Fruit trees are major cash crops that help to repurpose or reclaim barren agricultural lands and remediate soil and forest ecosystems. Breeding is vital to the development of the fruit tree industry. Accessions are original sources for breeding new fruit tree varieties and serve as the basis for studying the origin and evolution of tree species. Fruit tree accessions harbor numerous valuable genes [20]. In recent years, many countries have focused on the collection and preservation of plant genetic resources. Accession evaluation is of great practical significance in the maintenance of species diversity and is now a topic of global concern.

In previous studies, researchers have measured and analyzed the fruit quality of *P. humilis* accessions. Total flavonoid content and radical scavenging activity of 16 *P. humilis* genotypes have been determined [3]. The polyphenol compounds of 28 and 13 different genotypes of *P. humilis* in Liaoning province [21] and Beijing city [2] have been systematically characterized, respectively. Seven varieties of *P. humilis* have been evaluated [22]. Although the fruit quality of *P. humilis* has been evaluated, the number of accessions was small. To the best of our knowledge, this is the first study to assess the fruits derived from *P. humilis* accessions on a large scale. We selected 137 *P. humilis* accessions and analyzed the essential characteristics, bioactive compounds, and antioxidant capacities of the fruit to evaluate the fruit quality in *P. humilis* and identify cultivars that could provide superior human health benefits. Such investigations may help in promoting the use of *P. humilis* in functional foods and as an ingredient in pharmaceutical and nutraceutical products.

## Materials and methods

### Plant materials and chemicals

The experimental materials were acquired from the horticultural station, the accession nursery, and the Juxin experimental orchard of Shanxi Agricultural University, Jinzhong, China

(37˚23′N, 112˚29′E). Field management included conventional irrigation and fertilization methods. During the growth period, the plants were fertilized twice a year, once before germination in spring and once after winter. The amount was 40 kg of nitrogen fertilizer per 667m$^2$, mainly by hole application. From May to August, the plants were watered once or twice a month, and, in early November the plants were irrigated with enough water for wintering. Between June and October, in 2018 and 2019, 137 fully ripe *P. humilis* accessions (S1 Table) were harvested. Three healthy plants were selected per accession. Thirty pest- and disease-free fruits were sampled from the top, middle, and bottom of each of the three sampled plants and stored at -40˚C before analysis.

Methanol and acetonitrile used in ultrahigh performance liquid chromatography (UHPLC) were purchased from OmniGene LLC (Morrisville, NC, USA). Analytical flavonoid standards, i.e., catechin, epicatechin, rutin, liquiritigenin, cyanidin-3-O-glucoside, gallic acid, Trolox, 1,1-diphenyl-2-picrylhydrazyl (DPPH), 2,2'-azino-bis(3-ethylbenzothiazoline-6-sulfonic acid (ABTS), quercetin-7-O-β-D-glucopyranoside, and 2,4,6-tri-(2-pyridyl)-1,3,5-triazine (TPTZ) were purchased from Solarbio Technology Co. Ltd. (Beijing, China).

## Single fruit and stone weights and fruit diameter

Fruit and stone weights were measured using an electronic balance (JJ224BC Changshu Shuangjie Instrument Co. Ltd., Changshu, China) with an accuracy of 0.0001 g. The vertical and horizontal diameters of the fruit were measured with an electronic digital caliper (Guilin Guanglu Digital Dynamometer Co. Ltd., Guilin, China).

## Fruit color

Peel color was sensorially evaluated by a team of experienced students and teachers. Thereafter, peel color parameters were measured with a spectrocolorimeter (YS3060 Shenzhen San'enshi technology Co. Ltd., Shenzhen, China) using five representative fruits. Color parameters were measured at the intersection of the fruit equator and suture line, at a single point per 90˚ rotation along the equator. Therefore, the color parameters were measured at four points per fruit, the average was calculated, and the means of the five fruits were averaged. The following color indices were used to determine fruit color.

**Soluble solid content (SSC).** The SSC was measured with a handheld refractometer (LH-T32 Hangzhou Luheng Technologies Co. Ltd., Hangzhou, China).

**Total flavonoid and phenol content.** The extracts were prepared according to the methods of Bai [23] and Guo et al. [24]. All the sampled fruits were pulverized in liquid nitrogen and extracted with 40% (v/v) acidified methanol. The ratio of material to solution was 1:10. The suspension was mixed by whirlpool oscillation, extracted by ultrasound at 40 kHz for 30 min, and centrifuged at 10 000 × *g* for 15 min at 4˚C. The extraction was repeated thrice, and the filtrates were pooled.

Total flavonoid content (TFC) was determined by NaNO$_2$-Al(NO$_3$)$_3$ colorimetry [24]. First, 0.8 mL of the extracted solution was transferred to a 10 mL volumetric flask containing 0.3 mL of 5% (w/v) NaNO$_2$. The mixture was shaken well and kept in the dark for 6 min. Thereafter, 0.3 mL of 10% (w/v) Al(NO$_3$)$_3$ was added to the mixture, shaken well, and kept in the dark for another 6 min. Then, 4 mL of 4% (w/v) NaOH was added to the mixture and shaken well. The volume was adjusted to 10 mL using 40% (v/v) methanol, and the mixture was shaken well and kept in the dark for 10 min. Finally, the absorbance was read at 510 nm in a spectrophotometer (UV-5200 Shanghai Yuanxi Instrument Co. Ltd., Shanghai, China) calibrated with a rutin standard curve.

Total phenol content (TPC) was determined by Folin-Ciocalteu colorimetry [25]. First, 0.2 mL of the extracted solution was added to a 10 mL volumetric flask containing 0.2 mL Folin-Ciocalteu reagent. The mixture was shaken well and left to stand for 4 min. Then, 1 mL of 10% (w/v) $Na_2CO_3$ was added to the mixture, and the volume was adjusted to 8 mL with $ddH_2O$. The mixture was shaken well and incubated in a water bath at 35°C for 1 h. The absorbance was read at 760 nm in a spectrophotometer (UV-5200 Shanghai Yuanxi Instrument Co. Ltd., Shanghai, China) calibrated using a gallic acid standard curve.

### Antioxidant capacity

**DPPH assay.** Following the method of Zhang [26], we added 2.8 mL of 0.1 mM DPPH to 0.2 mL of extracted solution. The mixture was shaken well and kept in the dark for 30 min at 25°C. For the blank, the extracted solution was replaced with 40% (v/v) methanol. The absorbance was read at 517 nm in a spectrophotometer (UV-5200 Shanghai Yuanxi Instrument Co. Ltd., Shanghai, China).

**ABTS assay.** Following the method of Zhang [26], we added 3.9 mL of $ABTS^+$ solution (7 mL ABTS plus 140 mM $K_2(SO_4)$) to 0.1 mL of extracted solution. The $ABTS^+$ solution was prepared in the dark at 25°C over 12–16 h. The sample mixture was shaken well and incubated in the dark for 10 min at room temperature. For the blank, the extracted solution was replaced with 40% (v/v) methanol. The absorbance was read at 734 nm in a spectrophotometer (UV-5200 Shanghai Yuanxi Instrument Co. Ltd., Shanghai, China).

**Ferric reducing antioxidant power (FRAP) assay.** Following the method of Zhang [26], we added 4.9 mL of FRAP solution (0.1 M $CH_3COONa$ [pH = 3.6], 10 mM TPTZ, and 20 mM $FeCl_3$ in a 10:1:1 volumetric ratio) to 0.1 mL of extracted solution. The sample mixture was shaken well and incubated in the dark for 10 min at room temperature. For the blank, the extracted solution was replaced with 40% (v/v) methanol. The absorbance was read at 593 nm in a spectrophotometer (UV-5200 Shanghai Yuanxi Instrument Co. Ltd., Shanghai, China).

**Trolox evaluation.** The standard antioxidant Trolox was used to plot a standard curve and evaluate antioxidant activity [27]. The results of DPPH, FRAP and ABTS were expressed as mg of Trolox equivalents (TE) per g of fruit (on a fresh weight basis)

### Flavonoid components

The content of six flavonoid components was determined by UHPLC in an Agela Venusil ABS C18 column (4.6 mm × 250 mm; 5 μm) (Agela Technologies, Wilmington, DE, USA). In particular, 1 mL of flavonoid extract was passed through a 0.22 μm Millipore membrane filter (EMD Millipore, Burlington, MA, USA) and placed in a liquid sample bottle. The solvent system consisted of 0.5% (v/v) formic acid water (solvent A) and acetonitrile (solvent B). The flow rate was set to 0.8 mL min$^{-1}$, and the run time was 69 min. The sample injection volume was 20 μL. The gradient program was as follows: 10% B at 0 min; 13% B for 0–5 min; 16% B for 5–25 min; 21% B for 25–30 min; 22% B for 30–45 min; 25% B for 45–50 min; 25% B for 50–65 min; and 10% B for 65–69 min. The detector was set to 280 nm (for detecting catechin, epicatechin, and liquiritigenin), 360 nm (rutin and quercetin-7-O-β-D-glucopyranoside), and 520 nm (cyanidin-3-O-glucoside) for the simultaneous monitoring of the various flavonoid components.

### Statistical analysis

Data were analyzed in Microsoft Excel v. 2007 (Microsoft Corporation, Redmond, WA, USA). Cluster analysis, principal component analysis and differences among the mean values were evaluated using ANOM (analysis of means), and statistical significance was set at $P < 0.05$

**Table 1. Flavonoid content and variation analysis of Prunus humilis fruits in 2018 and 2019.**

|  | The average of total flavonoid content (mg/g FW) |
|---|---|
| 2018 | 11.11 |
| 2019 | 10.58 |
| Coefficient of variation (%) | 12.43 |

using the Statistical Analysis System v. 9.2 (SAS Institute, Cary, NC, USA). The coefficient of variation (CV) was calculated as CV (%) = S/F × 100 (SAS Institute, Cary, NC, USA), where S and F are the standard deviation and average, respectively. The abbreviations are detailed in S2 Table.

# Results

## Changes in flavonoid content over two years

The flavonoid content and the antioxidant capacities could be greatly affected by climatic conditions, we collected two-year data for analysis. In 2018 and 2019, the average contents of flavonoid were 11.11 mg/g FW and 10.58 mg/g FW, respectively (Table 1), with a difference between two years was of 0.53 mg/g. The average variation coefficient of flavonoid contents in the same P. humilis accession was 12.43%, less than 20%. The results showed that the change in the flavonoid content of the same P. humilis accession was small and the flavonoid content was relatively stable between the two years.

## Fruit character

The average single fruit and stone weights for 137 *P. humilis* accessions were 6.13 g and 0.452 g, respectively, with a CV of 45.98% and 38.02%, respectively (Table 2). As both had a CV >20%, the fruit and stone weights showed large variation and rich genetic diversity. The average vertical and horizontal diameters were 19.62 mm and 21.88 mm, respectively, with a CV of 15.52% and 17.19%, respectively. As both had a CV <20%, the vertical and horizontal diameters showed small variation, and their inheritance was relatively simple. The average vertical diameter was smaller than the average horizontal diameter, which is consistent with the oblate shape found in most *P. humilis* fruits. The average SSC was 14.15%, and the CV was 17.71%. Hence, the variation in SSC was small, and the inheritance relatively simple.

The average TFC and TPC were 10.58 mg/g FW and 3.93 mg/g FW, respectively, with a CV of 35.78% and 34.06%, respectively (Table 3). Therefore, both had a CV >20%, and as we found significant differences among the accessions ($P < 0.01$), this indicated a large variation in the TFC and TPC of the *P. humilis* accessions and a rich genetic diversity. The highest average ABTS scavenging ability was 12.38 mg (TE)/g FW. The average of FRAP scavenging ability was 8.03 mg (TE)/g FW. The lowest average DPPH scavenging ability was 4.42 mg (TE)/g FW. The CV for these antioxidant indices was >20%, and we found significant differences among

**Table 2. Fruit character variation in 137 *Prunus humilis* accessions.**

| Fruit trait | Average | Maximum | Minimum | Standard deviation | Coefficient of variation (%) |
|---|---|---|---|---|---|
| Single fruit weight (g) | 6.13 | 15.61 | 1.20 | 2.82 | 45.98 |
| Stone weight (g) | 0.452 | 0.954 | 0.174 | 0.17 | 38.02 |
| Vertical diameter (mm) | 19.62 | 26.41 | 12.13 | 3.04 | 15.52 |
| Horizontal diameter (mm) | 21.88 | 31.32 | 12.84 | 3.76 | 17.19 |
| Soluble solid content (%) | 14.15 | 20.64 | 7.04 | 2.51 | 17.71 |

**Table 3. Fruit bioactive compounds and antioxidant capacities in 137 *Prunus humilis* accessions.**

| Fruit trait | Average | Maximum | Minimum | Standard deviation | Coefficient of variation (%) | *F*-value |
|---|---|---|---|---|---|---|
| TFC (mg/g FW) | 10.58 | 28.37 | 3.90 | 3.78 | 35.78 | 94.08** |
| TPC (mg/g FW) | 3.93 | 9.02 | 1.44 | 1.34 | 34.06 | 99.80** |
| DPPH (mg TE/g FW) | 4.42 | 8.19 | 2.11 | 1.14 | 25.86 | 58.11** |
| FRAP (mg TE/g FW) | 8.03 | 16.82 | 3.25 | 2.56 | 31.88 | 69.50** |
| ABTS (mg TE/g FW) | 12.38 | 24.23 | 4.68 | 3.88 | 31.32 | 35.29** |

Note: *F*-value and **indicates significance at $P < 0.01$. TFC, total flavonoid content; TPC, total phenol content; DPPH, 2,2-diphenyl-1-picrylhydrazyl free radical scavenging capacity; FRAP, ferric reducing antioxidant power; ABTS, 2,2'-azinobis (3-ethylbenzothiazoline-6-sulfonic acid) free radical scavenging capacity.

accessions ($P < 0.01$). Thus, the antioxidant capacities of the *P. humilis* accessions varied greatly and had rich genetic diversity.

## Cluster analysis of fruit flavonoid content

In the cluster analysis, the accessions were divided according to flavonoid content into four major types and six subgroups (S1 Fig; Table 4). One accession with ultrahigh flavonoid content (28.37 mg/g FW) was assigned to subgroup one. Ten accessions with high flavonoid content (range 15.96–21.99 mg/g FW; average 18.12 mg/g FW) were assigned to subgroup two. Twenty-two accessions with medium-high flavonoid content (range 12.91–15.23 mg/g FW; average 14.07 mg/g FW) were assigned to subgroup three. Forty-two accessions with medium flavonoid content (range 9.57–12.84 mg/g FW; average 11.06 mg/g FW) were assigned to subgroup four. Forty accessions with medium-low flavonoid content (range 7.21–9.40 mg/g FW; average 8.32 mg/g FW) were assigned to subgroup five. Twenty-two accessions with low flavonoid content (range 3.90–6.92 mg/g FW; average 6.03 mg/g FW) were assigned to subgroup six. Overall, we found that the flavonoid content of the 137 accessions was normally distributed (Fig 1) and mainly occurred at medium concentrations in the fruit (Table 4).

## Flavonoid content in different-colored fruit peels

We divided the 137 accessions into different peel colored phenotypes, i.e., dark red, red, light red, red-orange, and yellow (Fig 2). We then classified the flavonoid content for each phenotype into subgroups through cluster analysis (Table 5). The flavonoid content in the dark red accession was concentrated in the medium and medium-low subgroups (both with 40%). The flavonoid content in the red accession was distributed across all six subgroups but mainly occurred in the medium- and medium-low subgroups (28.40% and 27.16%, respectively). The

**Table 4. Flavonoid content levels in *Prunus humilis* accessions.**

| Flavonoid content types | Subgroup | Number/proportion of accessions (%) | No. | Flavonoid content range (mg/g FW) | Average flavonoid content (mg/g FW) |
|---|---|---|---|---|---|
| Ultrahigh | | 1/0.73 | 1 | 28.37 | 28.37 |
| High | | 10/7.30 | 2–11 | 15.96–21.99 | 18.12 |
| Medium | Medium-high | 22/16.06 | 12–31, 129–130 | 12.91–15.23 | 14.07 |
| | Medium | 42/30.66 | 32–65, 126–128, 131–135 | 9.57–12.84 | 11.06 |
| Low | Medium-low | 40/29.20 | 66–105 | 7.21–9.40 | 8.32 |
| | Low | 22/16.06 | 106–125, 136, 137 | 3.90–6.92 | 6.03 |

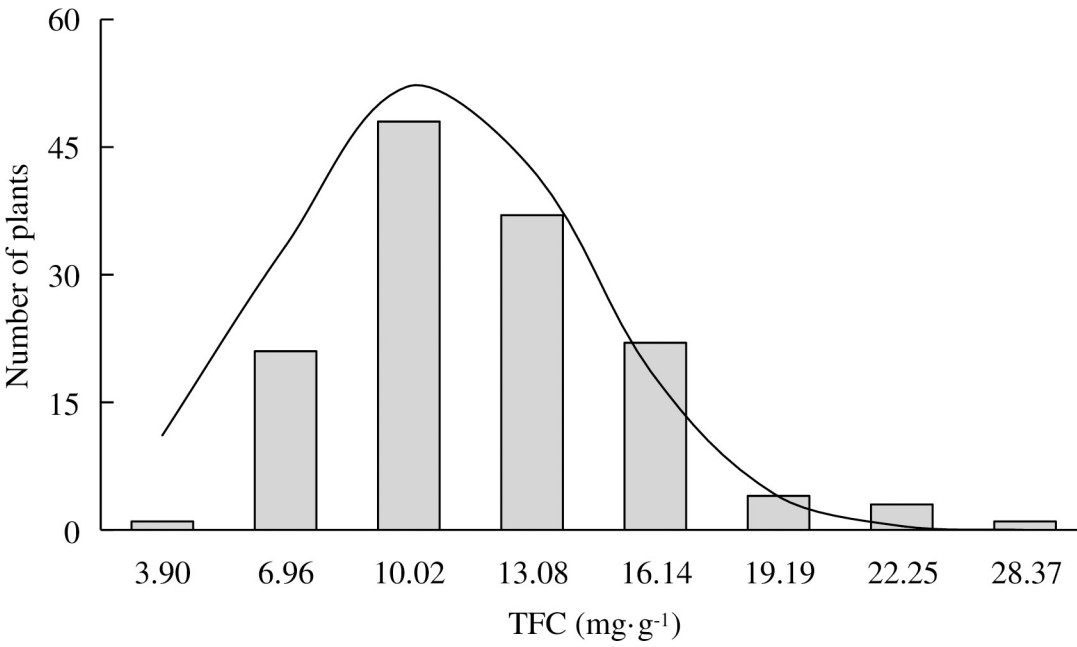

**Fig 1. Normal distribution of total flavonoid content in 137 *Prunus humilis* accessions.**

flavonoid content in the light red accession was distributed across the high, medium, medium-low, and low subgroups. However, the highest proportion occurred in the medium-low subgroup (42.86%). The flavonoid content of the red-orange and yellow accession varied widely across the subgroups, except in the ultrahigh subgroup. The highest flavonoid content in the red-orange and yellow accession occurred in the medium (41.67%) and medium-low (40%) subgroups, respectively.

## Bioactive compounds and antioxidant capacity

Based on the flavonoid cluster analysis and phenotype classification, we selected 62 of the 137 accessions for further analysis. The TFC and TPC of 62 accessions were 5.07–28.37 mg/g FW and 2.02–9.02 mg/g FW, respectively, and comprised all six subgroups (Fig 3). The flavonoid content was higher than the TPC in each accession.

Fig 4 shows the antioxidant capacities of 62 accessions. They most strongly scavenged ABTS (6.55–24.23 mg (TE)/g FW) followed by FRAP (3.82–16.82 mg (TE)/g FW) and DPPH (2.47–8.19 mg (TE)/g FW).

## Flavonoid components

The content of six flavonoid components were determined using analytical flavonoid standards (Fig 5). There were significant differences in the content of the flavonoid components among the 62 accessions (Table 6). The content of the flavonoid components differed significantly among the five different phenotypes ($P < 0.01$). Catechin was the only flavonoid component detected in all 62 accessions (15.76–120.81 mg/100 g FW). Epicatechin was detected in all accessions, except 628, DS-1, S-D-1, and 19–05 (1.98–20.47 mg/100 g FW). Liquiritigenin was detected in 54 accessions (1.05–1.95 mg/100 g FW). Its content and variation were the lowest among all six flavonoid components. Cyanidin-3-O-glucoside was detected in 39 accessions (15.24–231.18 mg/100 g FW). Cyanidin-3-O-glucoside was not detected in 23 accessions

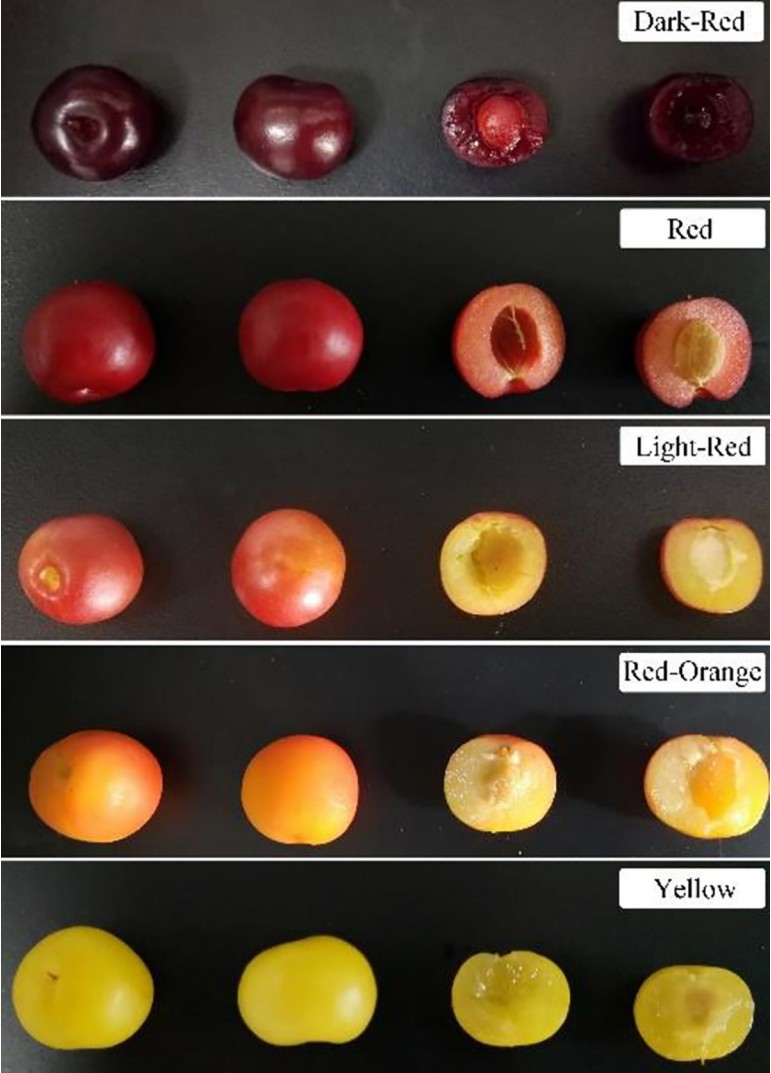

**Fig 2. *Prunus humilis* accession phenotypes.**

include all the yellow accessions, 80% of the light red accessions and 84.62% of the red-orange accessions, but it was detected in all red and dark red accessions. Overall, the color index increased with cyanidin-3-O-glucoside content. Thus, there appeared to be a correlation between cyanidin-3-O-glucoside and red peel color. Rutin was detected in all accessions,

**Table 5. Flavonoid content in different-colored fruit peels of *Prunus humilis* accessions.**

| Peel color | Flavonoid content | | | | | |
|---|---|---|---|---|---|---|
| | Ultrahigh (%) | High (%) | Medium-high (%) | Medium (%) | Medium-low (%) | Low (%) |
| Dark red | | | | 40.00 | 40.00 | 20.00 |
| Red | 1.23 | 6.17 | 18.52 | 28.40 | 27.16 | 18.52 |
| Light red | | 14.29 | | 28.57 | 42.86 | 14.29 |
| Red-orange | | 8.33 | 25.00 | 41.67 | 20.83 | 4.17 |
| Yellow | | 10.00 | 5.00 | 25.00 | 40.00 | 20.00 |

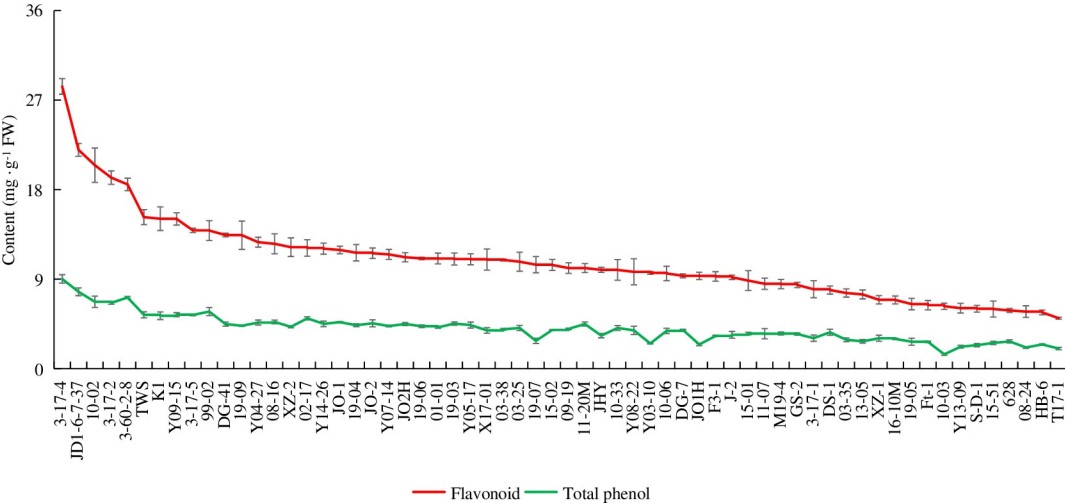

**Fig 3. Flavonoid and total phenol content in 62 *Prunus humilis* accessions.**

except Y13-09, Ft-1, T17-1, and 03–35 (1.57–3.25 mg/100 g FW). Quercetin-7-O-β-D-gluco-pyranoside was detected in all accessions, except Ft-1 and T17-1 (2.39–4.65 mg/100 g FW).

## Correlations among fruit color parameters, bioactive compound content, flavonoid components, and antioxidant capacities

A correlation analysis between the color parameters, bioactive compound content, flavonoid components, and antioxidant capacities of 62 *P. humilis* fruit accessions revealed that the color index was significantly positively correlated with cyanidin-3-O-glucoside content (correlation coefficient = 0.85; $P < 0.01$) (Fig 6). Catechin, epicatechin, and rutin content was significantly

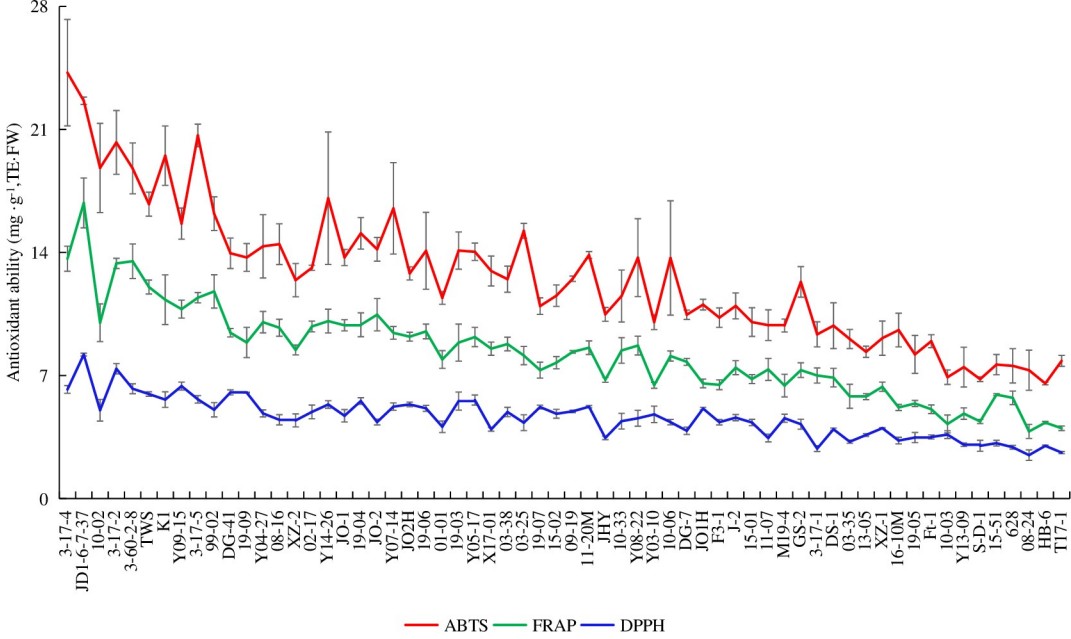

**Fig 4. Antioxidant capacities of 62 *Prunus humilis* accessions.**

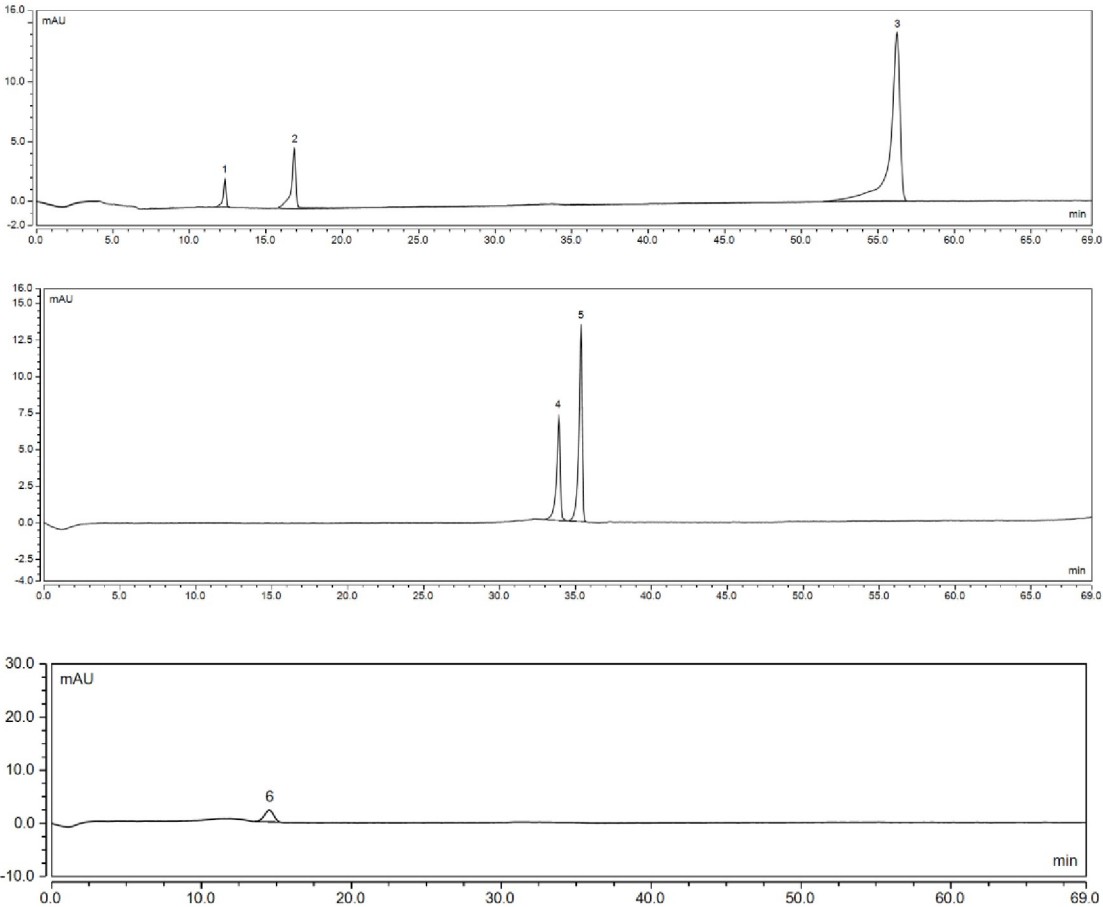

**Fig 5. Flavonoid standards used to detect flavonoid content in 62 *Prunus humilis* accessions by ultrahigh performance liquid chromatography (UHPLC). (a)** Catechin (1), epicatechin (2), and liquiritigenin (3) detected at 280 nm; **(b)** shows rutin (4) and quercetin-7-O-β-D-glucopyranoside (5) detected at 360 nm; **(c)** cyanidin-3-O-glucoside (6) detected at 520 nm.

positively correlated with TFC ($P < 0.01$). The strongest correlation was found between catechin and TFC (correlation coefficient = 0.80). TFC and TPC were significant positively correlated with the antioxidant indices ($P < 0.01$). Correlation coefficients between TFC and ABTS and between TPC and FRAP were 0.92 and 0.94, respectively.

## Principal component analysis

The PCA results identified two components that explained 86.12% of the total variation in fruit quality among the 62 *P. humilis* accessions (Table 7). The first principal component (Prin1) contributed 52.56% of the total variation, and large positive values were associated with total flavonoid content and ABTS, suggesting that these two indices contributed significantly to Prin1. The second principal component (Prin2) contributed 33.56% of the total variation; large positive values were associated with single fruit weight,vertical and horizontal diameter, suggesting that these three indices greatly contributed to Prin2.

Scatterplots from the PCA based on the fruit quality showed that 10–02, 3-17-2, 3-17-4, and JD1-6-7-37 belonged in the first group (Fig 7), characterized by high TFC, ABTS and large fruit. The other four groups had different characteristics. These data suggest that crossing within groups maybe more efficient for directed breeding.

**Table 6. Flavonoid components in different-colored fruit peels of 62 *Prunus humilis* accessions.**

| Accession name | Peel color | Color index | CC (mg/100 g FW) | EC (mg/100 g FW) | LG (mg/100 g FW) | C3G (mg/100 g FW) | RT (mg/100 g FW) | Q3G (mg/100 g FW) |
|---|---|---|---|---|---|---|---|---|
| 3-17-4 | Red | 188.71 | 81.12±2.51 | 9.11±1.11 | 1.52±0.01 | 23.47±0.15 | 1.95±0.02 | 2.63±0.04 |
| JD1-6-7-37 | Red | 75.01 | 120.81±5.41 | 15.32±1.32 | 1.28±0.02 | 31.38±0.71 | 1.95±0.01 | 3.43±0.05 |
| 10–02 | Red-orange | 24.13 | 53.64±2.29 | 3.03±0.26 | 1.61±0.08 | - | 2.13±0.08 | 2.90±0.04 |
| 3-17-2 | Red | 93.40 | 116.43±3.82 | 11.73±0.91 | 1.60±0.07 | 24.02±0.45 | 2.55±0.09 | 3.45±0.08 |
| 3-60-2-8 | Light red | 38.96 | 64.47±1.21 | 10.12±0.54 | 1.24±0.04 | - | 2.89±0.11 | 4.65±0.15 |
| TWS | Red | 112.08 | 84.87±1.98 | 8.53±1.52 | 1.67±0.03 | 47.73±0.60 | 2.14±0.03 | 3.08±0.01 |
| K1 | Red | 79.88 | 119.89±4.96 | 20.47±3.46 | 1.59±0.02 | 17.36±0.13 | 2.26±0.03 | 3.26±0.01 |
| Y09-15 | Red | 54.01 | 81.37±1.55 | 9.02±0.98 | 1.28±0.02 | 36.20±0.74 | 2.30±0.04 | 3.62±0.02 |
| 3-17-5 | Red-orange | 15.69 | 54.68±4.44 | 4.06±0.87 | 1.12±0.03 | - | 2.20±0.08 | 2.85±0.03 |
| 99–02 | Red-orange | 12.04 | 68.67±3.23 | 5.15±0.05 | 1.68±0.04 | - | 2.22±0.01 | 3.66±0.10 |
| DG-41 | Red | 60.71 | 83.82±2.24 | 3.48±0.42 | 1.41±0.02 | 27.75±0.94 | 2.09±0.02 | 3.17±0.04 |
| 19–09 | Red-orange | 28.17 | 60.12±0.86 | 14.17±1.01 | 1.22±0.02 | - | 1.89±0.01 | 2.80±0.03 |
| Y04-27 | Yellow | 5.15 | 65.30±3.26 | 2.99±0.10 | 1.16±0.01 | - | 2.43±0.05 | 4.29±0.13 |
| 08–16 | Yellow | 1.88 | 68.00±2.14 | 4.41±0.21 | 1.29±0.01 | - | 2.05±0.05 | 2.79±0.02 |
| XZ-2 | Light red | 74.25 | 52.55±2.21 | 2.69±0.01 | 1.21±0.03 | - | 1.97±0.02 | 2.81±0.01 |
| 02–17 | Light red | 11.66 | 39.64±2.15 | 2.96±0.28 | 1.25±0.02 | - | 1.91±0.03 | 2.94±0.15 |
| Y14-26 | Red | 90.02 | 49.92±0.57 | 4.49±0.34 | 1.25±0.02 | 21.18±0.63 | 1.60±0.01 | 2.45±0.01 |
| JO-1 | Dark red | 454.33 | 45.61±2.00 | 5.97±0.26 | 1.29±0.02 | 139.78±1.91 | 2.10±0.02 | 3.46±0.07 |
| 19–04 | Red | 67.29 | 52.26±2.42 | 7.46±0.78 | 1.28±0.03 | 43.5±2.50 | 1.92±0.06 | 3.93±0.13 |
| JO-2 | Dark red | 576.91 | 57.46±4.57 | 9.17±1.13 | 1.33±0.03 | 143.11±4.51 | 2.41±0.06 | 3.52±0.15 |
| Y07-14 | Red-orange | 7.03 | 45.83±1.16 | 2.75±0.07 | 1.49±0.01 | 15.24±0.01 | 1.95±0.08 | 2.97±0.04 |
| JO2H | Red | 124.17 | 51.56±4.48 | 3.40±0.30 | 1.37±0.04 | 68.81±2.07 | 2.27±0.02 | 3.55±0.05 |
| 19–06 | Red | 97.38 | 38.85±1.60 | 2.12±0.07 | 1.19±0.01 | 38.51±1.63 | 1.87±0.02 | 2.58±0.02 |
| 01–01 | Red-orange | 15.68 | 52.36±2.20 | 5.33±0.32 | 1.63±0.03 | - | 2.00±0.01 | 3.31±0.04 |
| 19–03 | Red-orange | 16.43 | 63.88±5.67 | 3.37±0.08 | 1.95±0.04 | - | 1.94±0.05 | 3.44±0.17 |
| Y05-17 | Red-orange | 25.75 | 59.85±1.96 | 6.26±0.63 | 1.13±0.01 | - | 1.73±0.05 | 2.71±0.01 |
| X17-01 | Red | 36.79 | 39.81±4.63 | 3.25±0.12 | 1.58±0.04 | 20.63±2.94 | 2.08±0.03 | 2.89±0.03 |
| 03–38 | Red-orange | 8.84 | 33.66±0.57 | 2.94±0.16 | 1.39±0.02 | - | 1.71±0.04 | 2.54±0.06 |
| 03–25 | Red | 256.74 | 60.25±0.64 | 5.32±0.25 | 1.28±0.01 | 58.67±1.81 | 2.09±0.05 | 3.26±0.10 |
| 19–07 | Red-orange | 11.73 | 37.96±3.91 | 3.66±0.82 | 1.05±0.02 | 23.82±0.89 | 1.88±0.02 | 3.99±0.04 |
| 15–02 | Red | 53.33 | 31.57±2.31 | 3.64±0.28 | 1.17±0.01 | 20.89±1.79 | 1.92±0.01 | 3.01±0.04 |
| 09–19 | Yellow | 3.24 | 46.44±2.00 | 3.70±0.08 | 1.51±0.04 | - | 1.85±0.03 | 3.21±0.03 |
| 11-20M | Red | 83.14 | 35.85±2.76 | 4.99±0.43 | 1.38±0.03 | 15.9±1.27 | 3.25±0.02 | 3.05±0.12 |
| JHY | Red | 69.14 | 31.84±3.07 | 3.44±0.21 | 1.19±0.01 | 23.05±0.22 | 1.84±0.05 | 2.76±0.01 |
| 10–33 | Red | 108.92 | 29.61±0.87 | 3.25±0.31 | 1.10±0.03 | 42.45±3.37 | 1.89±0.06 | 2.65±0.05 |
| Y08-22 | Red | 59.99 | 34.03±2.73 | 2.05±0.11 | 1.09±0.03 | 34.87±1.88 | 1.63±0.04 | 2.66±0.04 |
| Y03-10 | Red | 72.73 | 31.01±1.61 | 3.51±0.07 | 1.11±0.01 | 45.51±1.09 | 2.17±0.03 | 3.10±0.08 |
| 10–06 | Yellow | 5.92 | 28.05±2.47 | 2.08±0.10 | 1.46±0.69 | - | 1.58±0.01 | 2.92±0.02 |
| DG-7 | Red | 66.92 | 36.75±1.94 | 3.01±0.38 | 1.36±0.01 | 17.93±0.04 | 2.22±0.01 | 3.12±0.06 |

(*Continued*)

**Table 6.** (Continued)

| Accession name | Peel color | Color index | CC (mg/100 g FW) | EC (mg/100 g FW) | LG (mg/100 g FW) | C3G (mg/100 g FW) | RT (mg/100 g FW) | Q3G (mg/100 g FW) |
|---|---|---|---|---|---|---|---|---|
| JO1H | Red | 90.12 | 42.69±3.95 | 2.93±0.02 | 1.35±0.03 | 28.93±1.02 | 2.08±0.03 | 2.81±0.03 |
| F3-1 | Red-orange | 47.75 | 39.46±1.01 | 2.84±0.08 | 1.19±0.02 | - | 1.83±0.04 | 2.70±0.01 |
| J-2 | Yellow | 5.58 | 38.43±1.78 | 2.26±0.01 | 1.29±0.03 | - | 2.00±0.06 | 3.98±0.09 |
| 15–01 | Red-orange | 22.34 | 36.34±1.07 | 2.82±0.07 | 1.54±0.04 | - | 1.99±0.03 | 2.86±0.04 |
| 11–07 | Dark red | 2802.48 | 18.69±1.38 | 4.11±0.60 | 1.43±0.03 | 231.18±8.98 | 1.70±0.01 | 3.92±0.09 |
| M19-4 | Red | 42.75 | 25.29±0.08 | 1.98±0.05 | 1.56±0.03 | 18.94±0.21 | 1.59±0.01 | 2.70±0.18 |
| GS-2 | Red | 92.73 | 27.43±0.63 | 3.72±0.12 | 1.21±0.01 | 38.69±1.37 | 1.75±0.03 | 2.62±0.05 |
| 3-17-1 | Red | 73.81 | 31.61±1.32 | 2.97±0.21 | 1.46±0.06 | 29.33±1.34 | 2.54±0.07 | 3.03±0.08 |
| DS-1 | Light red | 31.73 | 21.43±0.48 | - | 1.37±0.08 | 18.33±0.20 | 1.84±0.01 | 2.39±0.06 |
| 03–35 | Red | 109.86 | 28.37±1.23 | 3.23±0.05 | - | 33.55±1.12 | - | 2.67±0.03 |
| 13–05 | Red-orange | 36.04 | 32.70±0.97 | 2.55±0.03 | 1.22±0.02 | - | 1.57±0.04 | 2.63±0.05 |
| XZ-1 | Red | 163.45 | 28.97±1.40 | 3.56±0.93 | 1.55±0.01 | 17.87±0.90 | 1.67±0.04 | 3.32±0.03 |
| 16-10M | Light red | 45.82 | 28.56±2.85 | 2.10±0.04 | 1.63±0.09 | - | 1.75±0.10 | 2.68±0.06 |
| 19–05 | Yellow | 3.12 | 28.51±1.44 | - | - | - | 2.07±0.04 | 3.59±0.15 |
| Ft-1 | Red | 49.98 | 24.63±0.93 | 2.55±0.03 | - | 18.77±0.06 | - | - |
| 10–03 | Red | 106.06 | 24.86±1.60 | 2.72±0.12 | - | 32.68±1.57 | 1.77±0.02 | 3.30±0.12 |
| Y13-09 | Yellow | 7.04 | 18.74±2.01 | 2.24±0.08 | - | - | - | 2.62±0.04 |
| S-D-1 | Red | 81.85 | 24.18±0.76 | - | - | 22.94±0.58 | 1.86±0.02 | 2.86±0.04 |
| 15–51 | Red | 104.05 | 29.10±1.57 | 2.78±0.55 | 1.08±0.01 | 69.1±3.81 | 1.60±0.02 | 2.87±0.06 |
| 628 | Dark red | 2601.90 | 15.76±0.30 | - | 1.45±0.04 | 149.16±3.56 | 1.67±0.01 | 3.42±0.05 |
| 08–24 | Red | 98.57 | 17.49±0.04 | 2.16±0.04 | - | 16.03±0.08 | 1.60±0.01 | 2.50±0.01 |
| HB-6 | Yellow | 1.12 | 29.06±0.74 | 2.76±0.04 | 1.23±0.04 | - | 1.80±0.01 | 2.51±0.01 |
| T17-1 | Red | 69.61 | 26.83±0.64 | 2.61±0.5 | - | 20.42±0.24 | - | - |
| *F*-value | | | 281.63** | 83.57** | 11.38** | 1277.93** | 167.39** | 130.14** |

Note: *F*-value and ** indicate significance at $P < 0.01$. Data are means ± SD. CC, catechin; EC, epicatechin; LR, liquiritigenin; RT, rutin; Q3G, quercetin-7-O-β-D-glucopyranoside; C3G, cyanidin-3-O-glucoside.

## Discussion

In this study, we selected 137 *P. humilis* accessions and analyzed the basic characteristics, bioactive compounds, and antioxidant capacities of their fruit to elucidate flavonoid biosynthesis and identify cultivars that could provide superior human health benefits. Typically, consumers first notice fruit size and appearance. These morphological traits are also important commercial indicators of fruit quality and have become the focus of plant breeders. The range of single fruit weights was wide (1.2–15.61 g; average: 6.13 g). *Prunus humilis* fruits in Hebei province were measured, and it was found that the largest weight was 10.11 g and the smallest weight was 3.67 g [22]; however, we found that our *P. humilis* weight varied more widely. It is well known that fruit weight varies between areas, but under similar cultivation conditions in the same area, *P. humilis* also produces fruit of different weights. This finding means that fruit weight depends on accessions and not geographic areas. Moreover, the genetic diversity in our study was rich. Hence, these accessions furnish abundant material for the selection and breeding of fruits of the required size. SSC directly determines fruit flavor; usually, the higher the soluble solid content, the better the flavor and quality. Thus, the selection of *P. humilis* with

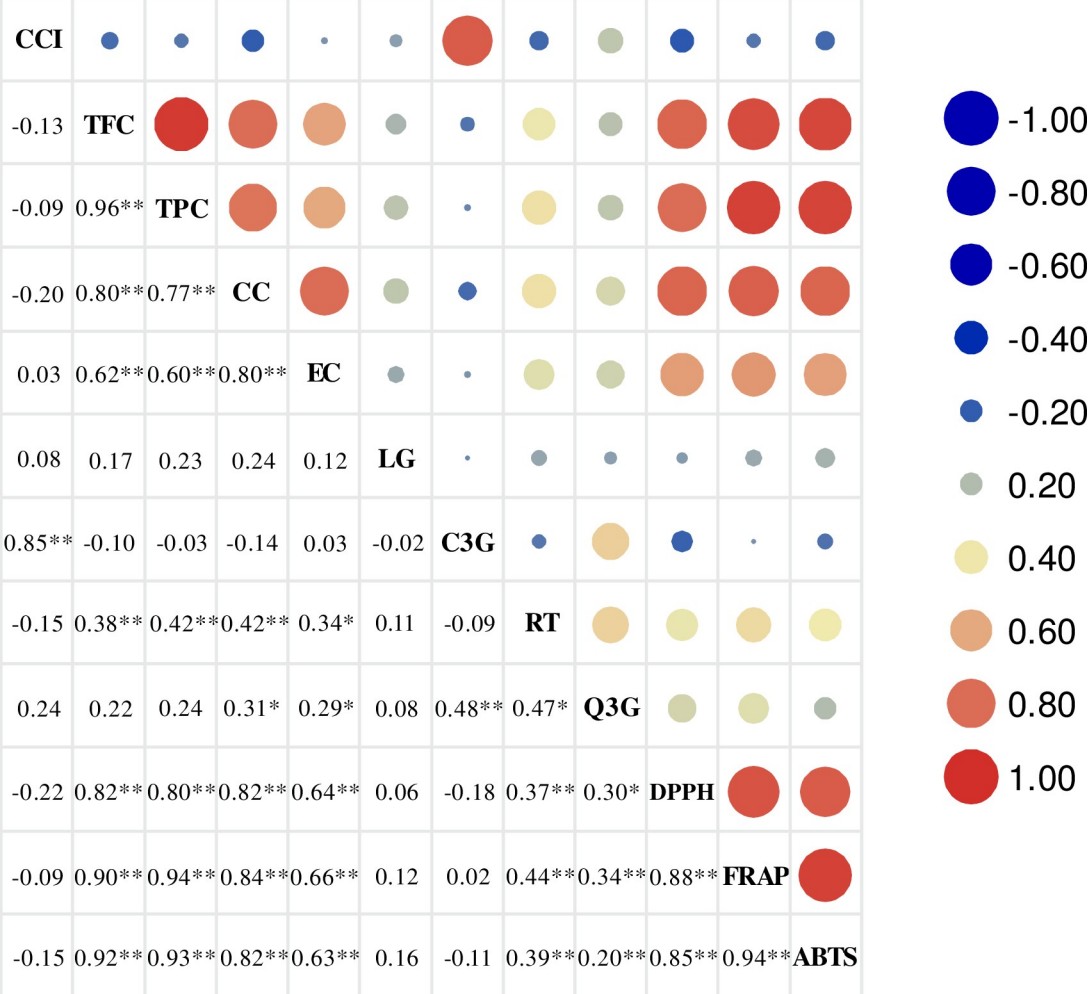

**Fig 6. Correlations among fruit color parameters, bioactive compound content, flavonoid components, and antioxidant capacities in 62 *Prunus humilis* accessions.** Note: * and ** indicate significance at $P < 0.05$ and $P < 0.01$, respectively. CC, catechin; EC, epicatechin; LR, liquiritigenin; RT, rutin; Q3G, quercetin-7-O-β-D-glucopyranoside; C3G, cyanidin-3-O-glucoside.

high SSC is an important breeding target. In this study, the SSC in *P. humilis* fruit was 7.04%–20.64%. Compared with the seven *P. humilis* fruits in Hebei province (7.43%-15.90%), our fruits have higher soluble solid content. Therefore, accessions with relatively higher SSC should be used in future breeding programs.

Flavonoids occur widely in plants and perform various physiological functions. Flavonoids are some of the most abundant polyphenols in the human diet [28] and are vital functional substances. Liu et al. [29] determined the TFC of five mulberry fruit varieties and found that the highest value was 1.46 mg/g and the lowest value was 0.86 mg/g. Wang et al. [30] measured the TFC for 13 strawberry fruit cultivars and reported a range of 0.26–0.54 mg/g. Xia et al. [31] evaluated the TFC of 13 *Prunus salicina* fruits and indicated a range of 0.14–1.44 mg/g and an average of 0.85 mg/g. Chen et al. [32] assessed the TFC of six apple fruit lines and found that the highest value was 4.40 mg/g, while the lowest value was 0.86 mg/g. Cao [33] determined the TFC of 186 pear accessions and showed that the values ranged between 0.19 mg/g and 6.77 mg/g and their average was 0.83 mg/g. Wu et al. [34] evaluated the TFC of five blueberry

**Table 7. Principal component analysis of fruit traits in 62 *Prunus humilis* accessions.**

| Traits | Eigenvectors | |
|---|---|---|
| | Prin1 | Prin2 |
| Single fruit weight (g) | -0.26 | 0.45 |
| Stone weight (g) | -0.01 | 0.03 |
| Vertical diameter (mm) | -0.20 | 0.48 |
| Horizontal diameter (mm) | -0.30 | 0.60 |
| Soluble solid content (%) | 0.10 | -0.05 |
| Total flavonoid content (mg/g FW) | 0.57 | 0.31 |
| Total phenol content (mg/g FW) | 0.19 | 0.08 |
| DPPH (mg TE/g FW) | 0.13 | 0.08 |
| FRAP (mg TE/g FW) | 0.35 | 0.14 |
| ABTS (mg TE/g FW) | 0.54 | 0.27 |
| Eigenvalue | 45.40 | 28.99 |
| Proportion (%) | 52.56 | 33.56 |
| Cumulative (%) | 52.56 | 86.12 |

Note: DPPH, 2,2-diphenyl-1-picrylhydrazyl free radical scavenging capacity; FRAP, ferric reducing antioxidant power; ABTS, 2,2'-azinobis (3-ethylbenzothiazoline-6-sulfonic acid) free radical scavenging capacity.

cultivars and demonstrated that the highest value was 3.06 mg/g and the lowest value was 0.52 mg/g. Jiang [35] measured the TFC of 50 grape materials including ten species (46 strains) of East Asian wild grapes; the range for the four control materials was 0.005–0.04 mg/g. In the present study, the TFC of 137 *P. humilis* accessions ranged from 3.90 mg/g FW to 28.37 mg/g FW, with an average of 10.58 mg/g. The polyphenol content of 28 different genotypes in

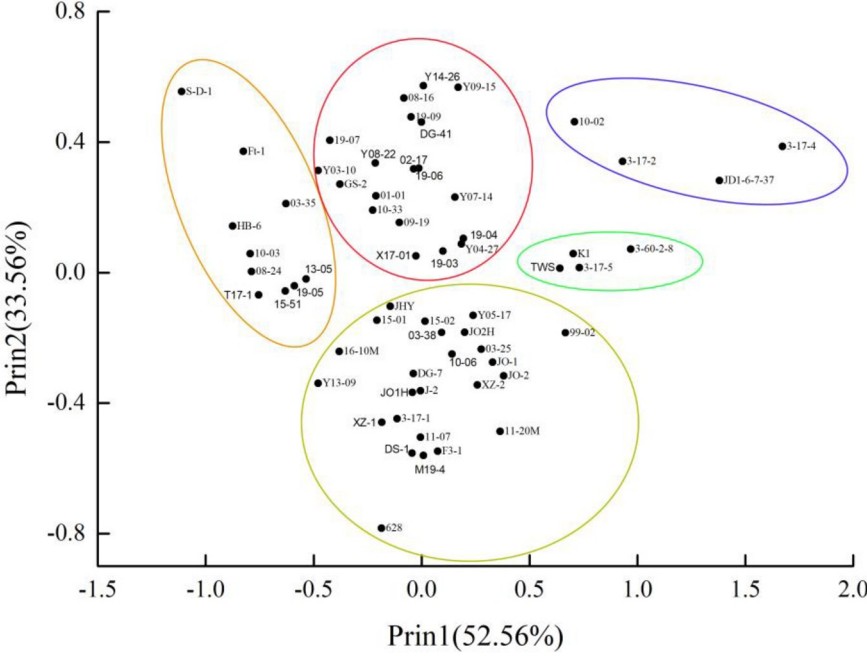

**Fig 7. Scatterplot of principal component analysis based on fruit quality of 62 *Prunus humilis* accessions.** The five circles indicate the accessions belonging to the top two principal components.

Liaoning Province was determined, and a total of 31 polyphenols was detected, including 17 flavonols and 1 flavone; among them, the Qu3Ara (quercetin 3-O-arabinoside) was the dominant component, and content ranging from 7.44 to 37.89 mg/100g DW [21]. Catechin was the only flavonoid component detected in all 62 accessions analyzed. Its content ranged from 15.76 to 120.81 mg/100 g FW, and it was significantly and positively correlated with TFC ($P < 0.01$; correlation coefficient = 0.80). Thus, catechin was the most important flavonoid component of all six components analyzed in this study. Catechin is a natural flavonoid in the flavanol family. It has a strong antioxidant capacity and scavenges free radicals in the human body. It can protect the heart and kidneys, normalize blood pressure, and prevent and cure cancer and inflammation [36]. Cyanidin-3-O-glucoside is a member of the anthocyanin family, which comprises the largest group of pigments in reddish fruits such as grapes, cherries, blueberries, blackberries, plums, and apples [37–39]. Cyanidin-3-O-glucoside is also the major anthocyanin in *P. humilis*. In the present study, cyanidin-3-O-glucoside was detected in all red and dark red *P. humilis* accessions and was absent in 23 other accessions, including in all the yellow, 80% of the light red, and 84.62% of the red-orange peels. Moreover, the color index increased with cyanidin-3-O-glucoside content, and the two parameters were significantly positively correlated ($P < 0.01$; correlation coefficient = 0.85). As we know red fruits and mainly dark red fruits have large quantities of these anthocyanidins. In our early research, the metabonomics of 19–04 (red) were determined, and 20 components of anthocyanins were detected. After absolute quantitative analysis by UHPLC, the content of cyanidin 3-O-glucoside accounted for more than 63% of the total content. Therefore, cyanidin-3-O-glucoside is a vital component of red peel color formation in *P. humilis* fruit.

Vitamins, organic acids, amino acids, phenols, flavonoids, superoxide dismutase, and other active substances in fruits play important roles in human antioxidant responses. They scavenge free radicals and participate in anti-aging, anti-radiation, and anticancer mechanisms [40]. These active substances also have strong free radical scavenging and antioxidant capacities in plant cells. Free radicals are unstable and therefore have a short life span. It is an important method to evaluate the antioxidant activity of antioxidants by studying their scavenging ability. Due to the difference of chemical property and reaction environment of different free radicals, it is important to select suitable free radicals for evaluating the biological activity and structure-activity relationship of free radical scavengers [1]. In the present study, DPPH, FRAP, and ABTS were selected as antioxidant indices. Regarding sweet cherry, the fruits were characterized by higher antioxidant activity, DPPH radical scavenging activity was about 10 mmol TE/100g DW [41], and that was strongly and positively correlated with fruit phenolic content [18]. The antioxidation test on *P. humilis* in Liaoning Province found that the FRAP free radical scavenging capacity (9.52–29.44mg/g DW) was the highest and the ABTS free radical scavenging capacity was the lowest (3.40–12.88mg/g DW) [21]. However, in our study, ABTS free radical scavenging capacity (4.68–24.23mg/g FW)was the highest; the difference may be due to fresh and dry samples, and the flavonoids may have changed during drying. This effect could cause a difference in free radical scavenging ability. Our results showed a significant positive correlation between flavonoid content and antioxidant capacity ($P < 0.01$). Although the *P. humilis* fruit is abundant in flavonoids, which act as potent antioxidants, the fruit might contain relatively more components that scavenge ABTS and fewer components that scavenge DPPH. We found that ABTS was the most scavenged, while DPPH was the least scavenged, and that the correlation coefficient for ABTS scavenging was the highest, while that for DPPH scavenging was the lowest. These findings are consistent with those reported for *Actinidia* spp. (kiwifruit) [42], *Lonicera caerulea* (honeyberry) [43], and *Rosa roxburghii* (chestnut rose) [44]. More than 9000 flavonoid components occur widely in plants. According to their structure, these flavonoids are classified as flavones, flavanones, flavanols, isoflavones, flavonols,

anthocyanidin, and flavanonols. Each component has its own function, and at the same time, the difference in flavonoid content causes differences in antioxidant activity. So different species differ in antioxidant capacity. In particular, *P. humilis* has a stronger ability of scavenging ABTS free radicals probably because contain a higher number of different antioxidant metabolites.

Cluster analysis divided the 137 *P. humilis* accessions by flavonoid content into four major types and six subgroups. The TFC in *P. humilis* accessions showed a normal distribution and was predominantly in the medium range (9.57–15.23 mg/g FW). These quantitative genetic traits are controlled by multiple genes. The accessions were also divided into five different peel color phenotypes. Fruits with red peel had the highest flavonoid content. When we integrated the flavonoid content phenotypes into the clustering results, we found that *P. humilis* accessions with different peel colors were distributed across all flavonoid concentrations. Hence, there was no obvious correlation between peel color and TFC.

The principal component analysis results identified two components, including TFC, ABTS free radical scavenging activity, single fruit weight,vertical and horizontal diameter. Based on these five indices, we grouped different accessions into five categories with different characteristics; 10–02, 3-17-2, 3-17-4 and JD1-6-7-37 are rich in TFC and strong antioxidant activity that could be more widely used in the general population and the food industry as a source of bioactives to improve human health.

## Conclusions

We explored the basic traits, bioactive compounds, and antioxidant capacities of *P. humilis* fruits. The single fruit and stone weights varied greatly and presented abundant genetic diversity. In contrast, the variation in SSC was small. Therefore, inheritance is relatively simple in this crop. The fruit shape was found to be basically oblate. We found significant differences among all accessions ($P < 0.01$) in terms of their TFC, TPC, and antioxidant capacity. These findings confirm that there is wide variation and rich genetic diversity among accessions. The TFC for most *P. humilis* accessions were in the medium range and were normally distributed. This quantitative genetic trait is controlled by multiple genes. Relatively more accessions had a red peel color than peels of other colors; however, we found no obvious correlation between peel color and TFC. The catechin content was high in the accessions and most strongly correlated with TFC. Therefore, catechin appears to be a vital flavonoid component in *P. humilis* fruit. Additionally, we found that cyanidin-3-O-glucoside was essential for peel color formation in red *P. humilis* fruits. We selected four accessions (10–02, 3-17-2, 3-17-4, JD1-6-7-37) with high TFC, ABTS free radical scavenging capacity, and large fruit.

## Supporting information

**S1 Fig. Cluster analysis of fruit flavonoid content in 137 *Prunus humilis* accessions.** (DOC)

**S1 Table. The background, maturing period and peel color of *Prunus humilis* accessions.** (DOC)

**S2 Table. A list of abbreviations.** (DOC)

**S3 Table.** (DOC)

## Acknowledgments

We are very grateful to the Wiley Editing for the assistance with language editing. We also thank the editors and reviewers for their helpful comments regarding this manuscript.

## Author Contributions

**Investigation:** Hongbo Fu, Baochun Fu.

**Methodology:** Pengfei Wang, Jiancheng Zhang, Junjie Du.

**Software:** Hongbo Fu.

**Writing – original draft:** Hongbo Fu.

**Writing – review & editing:** Xiaopeng Mu, Junjie Du.

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
