## [Editor Report · Decision Letter 0]

9 Oct 2020

PONE-D-20-31344

Fruit quality and antioxidant potential of Chinese dwarf cherry (Cerasus humilis) germplasms

PLOS ONE

Dear Dr. Fu,

Thank you for submitting your manuscript to PLOS ONE. After careful consideration, we feel that it has merit but does not fully meet PLOS ONE’s publication criteria as it currently stands. Therefore, we invite you to submit a revised version of the manuscript that addresses the points raised during the review process.

The manuscript preparation did not follow the Submission Guidelines for PLOS ONE. The authors are advised to meticulously chech and adapt the text according to the Submission Guidelines, since the manuscript cannot be sent to reviewers in this form.

We look forward to receiving your revised manuscript.

Kind regards,

Branislav T. Šiler, Ph.D.

Academic Editor

PLOS ONE

Journal Requirements:

2. Please include a copy of Table 4 which you refer to in your text on page 12. (Table 3 2x)
---

## [Author Response · Author response to Decision Letter 0]

22 Oct 2020

Dear editor:

All relevant data are within the manuscript and its Supporting Information files.And all available database are present as tables and figures.

Thanks!

---

## [Decision Letter · Decision Letter 1]

3 Nov 2020

PONE-D-20-31344R1

Fruit quality and antioxidant potential of Chinese dwarf cherry (Cerasus humilis) germplasms

PLOS ONE

Dear Dr. Fu,

Thank you for submitting your manuscript to PLOS ONE. After careful consideration, we feel that it has merit but does not fully meet PLOS ONE’s publication criteria as it currently stands. Therefore, we invite you to submit a revised version of the manuscript that addresses the points raised during the review process.

The authors show lack of familiarity with the study species since they did not perform literature survey in order to compare the results they have obtained with the ones already published, e.g.:

Shuang, C. H. E. N. G. (2007). Free Radical Scavenging Activities of Polyphenols from Prunus humilis Bge Fruit [J]. Food Science, 9.

Liu, S., Li, X., Guo, Z., Zhang, X., & Chang, X. (2018). Polyphenol content, physicochemical properties, enzymatic activity, anthocyanin profiles, and antioxidant capacity of Cerasus humilis (Bge.) Sok. Genotypes. Journal of Food Quality, 2018.

Wu, Q., Yuan, R. Y., Feng, C. Y., Li, S. S., & Wang, L. S. (2019). Analysis of polyphenols composition and antioxidant activity assessment of Chinese dwarf cherry (Cerasus humilis (bge.) Sok.). Natural Product Communications, 14(6), 1934578X19856509.

and other relevant articles.

*Cerasus humilis* (Bunge) S.Ya.Sokolov is a synonym of *Prunus humilis* Bunge (http://www.plantsoftheworldonline.org/taxon/urn:lsid:ipni.org:names:721943-1;
http://www.theplantlist.org/tpl1.1/record/rjp-33381)

and should be treated this way throughout the manuscript.

Main title: "accessions", not "germplasms" (and further in the text). Germplasm is a collective noun and cannot read in plural.

Many data are not provided in the M&M section, such as info on the accessions studied. Inconsistencies in measuring process are also notified by the reviewers, while the data visualization can be improved.

The whole text has to be thoroughly reorganized in order to keep the flow and improve readability. Some parts (particularly Introduction and Discussion sections) need to be fully rewritten as suggested in the reviewers' comments.

We look forward to receiving your revised manuscript.

Kind regards,

Branislav T. Šiler, Ph.D.

Academic Editor

PLOS ONE

Reviewers' comments:

Reviewer's Responses to Questions

**Comments to the Author**

1. If the authors have adequately addressed your comments raised in a previous round of review and you feel that this manuscript is now acceptable for publication, you may indicate that here to bypass the “Comments to the Author” section, enter your conflict of interest statement in the “Confidential to Editor” section, and submit your "Accept" recommendation.

Reviewer #1: (No Response)

Reviewer #2: (No Response)

2. Is the manuscript technically sound, and do the data support the conclusions?

Reviewer #1: No

Reviewer #2: Yes

3. Has the statistical analysis been performed appropriately and rigorously? 

Reviewer #1: No

Reviewer #2: Yes

4. Have the authors made all data underlying the findings in their manuscript fully available?

Reviewer #1: No

Reviewer #2: Yes

5. Is the manuscript presented in an intelligible fashion and written in standard English?

Reviewer #1: Yes

Reviewer #2: Yes

6. Review Comments to the Author

Reviewer #1: In this study authors analysed in 137 Cerasus humilis germplasms the fruit features, polyphenol and flavonoid contents, and the antioxidant activity of fruit extracts with the aim to identify cultivars with increased nutritional and nutraceutical properties.

The aim of the study is interesting, but the quality of the manuscript is poor.

The introduction is superficial, and more background is needed in order to understand the overall purpose of this study: authors have not considered the presence in literature of many papers dealing with antioxidant metabolites and antioxidant properties in cherry fruits (some of them in the links below).

https://www.ncbi.nlm.nih.gov/pmc/articles/PMC6912798/

https://www.sciencedirect.com/science/article/abs/pii/S096399691100007X

https://www.sciencedirect.com/science/article/abs/pii/S0308814617313158

Material and Methods

The methodology is missing important details rendering the reproducibility of this study impossible.

Authors deal with 137 C. humilis germplasms; however, they do not include any detail of the species or accessions in the Material and Methods section. A more detailed explanation of plant material must be included in this section, and a list of the 137 plant accessions should be added as supplementary table.

“Field management included conventional irrigation and fertilization methods”. This is an international journal and the conventional specifications for cultivation in China may be different from those of other Nations. Therefore, everything must be written in detail or referred to another publication in which the cultivation method has been already explained in detail.

“Total phenol content (TPC) was determined by Folin-Ciocalteu colorimetry [17]. A certain volume of extracted solution”: what do you mean with “a certain volume”? This is a scientific article, everything must be correctly defined.

“Following the method of Zhang [18], we added 2.8 mL of 0.1 mM DPPH to 0.2 mL of sample diluent.”: What do authors mean with “sample diluent”? Methanolic extract?

About the FRAP assay: “For the blank, the diluent was replaced with 40% (v/v) methanol.”: Do authors mean “40% (v/v) acidified methanol”?

The Trolox assay must be better specified.

About the cluster analysis, I do not understand why the heat map analysis has been performed using only one parameter and not including all the analyses parameters. An analysis like heat map or PCA represents a useful tool for the analysis of multivariate data that could offer an overview of the results and visualize the real differences between different accessions.

The discussion is too general and not very focused on the data obtained. Probably the fact that the entire set of data was not subjected to a multivariate analysis (heay map or PCA) did not allow the authors to fully understand the differences between the accessions analysed and the potential of their own study.

I think at the moment the manuscript requires an extensive revision, including missing details, a better data analysis and a deep re-writing of the introduction and discussion. Therefore, I reject the manuscript encouraging the authors to resubmit it after addressing all my concerns.

Reviewer #2: The paper entitled “Fruit quality and antioxidant potential of Chinese dwarf cherry (Cerasus humillis) germplasms¨ aims to analyse flavonoid, total phenol content and antioxidant potential of 137 dwarf cherry germplasms.

Article satisfy all PLOS ONE publication criteria.

The presented studies were well planned, and methodology is appropriate for the aim of study. Authors decided to use various methods. In order to determine antioxidant activity against free radicals the following methods used: DPPH, ABTS and FRAP. Additionally, statistical analysis of the studies was performed.

In my opinion the manuscript is well prepared and obtained results are interesting and novelty. The results are well analysed, and form of their presentation is good. References are adequate. The obtained results are interesting and can be interpreted as good direction for future studies.

Authors point out that they aimed to elucidate flavonoid biosynthesis and I do not agree with them. In my opinion they have studied the fruit quality, mainly related to flavonoids, but not their biosynthesis.

A list of abbreviations will be very helpful.

Define the abbreviations at Table 6 legend.

Have you studied other anthocyanidins apart from Cyanidin-3-gluscoside? Red fruits and mainly dark red fruits should have large quantities of these compounds.

Discussion is comprehensive and readable but can be improved. A deeper discussion relating data is needed.

May be a Principal component analysis of data should be interesting.

In my opinion the paper can be published in PLOS ONE after some revisions.

7. PLOS authors have the option to publish the peer review history of their article (what does this mean?). If published, this will include your full peer review and any attached files.

Reviewer #1: No

Reviewer #2: No

---

## [Author Response · Author response to Decision Letter 1]

6 Nov 2020

Dear editor:

1.We have revised the information about the author (Baochun Fu and Jinming Guo).

2.We have upload the figures(1-8) as the Item type 'Figure' files.

Thanks!

---

## [Editor Report · Decision Letter 2]

10 Nov 2020

PONE-D-20-31344R2

Fruit quality and antioxidant potential of Chinese dwarf cherry (Cerasus humilis ) accessions

PLOS ONE

Dear Dr. Fu,

Thank you for submitting your manuscript to PLOS ONE. After careful consideration, we feel that it has merit but does not fully meet PLOS ONE’s publication criteria as it currently stands. Therefore, we invite you to submit a revised version of the manuscript that addresses the points raised during the review process.

The authors rushly returned the revised manuscript failing to address all the concerns raised in the editor's and reviewers' reports. Moreover, in their response to reviewers, for the comments which need a thorough discussion, it is not sufficient to reply with one universal answer such as: "modification have been made as required...". The authors should express their opinion about each and every issue noted by the reviewer while pointing out to the exact place in the text (line numbers) where the changes were made. I strongly encourage the authors to meticulously check the previous editor's and reviewers' reports in order to avoid potential future flaws in the manusript preparation process.

We look forward to receiving your revised manuscript.

Kind regards,

Branislav T. Šiler, Ph.D.

Academic Editor

PLOS ONE

---

## [Author Response · Author response to Decision Letter 2]

12 Nov 2020

Dear editor:

We have upload the files about our manuscript.

Thanks!

---

## [Decision Letter · Decision Letter 3]

27 Nov 2020

PONE-D-20-31344R3

Fruit quality and antioxidant potential of Chinese dwarf cherry (Cerasus humilis ) accessions

PLOS ONE

Dear Dr. Fu,

Thank you for submitting your manuscript to PLOS ONE. After careful consideration, we feel that it has merit but does not fully meet PLOS ONE’s publication criteria as it currently stands. Therefore, we invite you to submit a revised version of the manuscript that addresses the points raised during the review process.

The language is still one of the major concerns. Some of them are stated in the reviewers' reports but I strongly recommend revising the manuscript by a professional editing agency.

Another major point: in the Revision #2, I advised:

"*Cerasus humilis* (Bunge) S.Ya.Sokolov is a synonym of *Prunus humilis* Bunge (http://www.plantsoftheworldonline.org/taxon/urn:lsid:ipni.org:names:721943-1;
http://www.theplantlist.org/tpl1.1/record/rjp-33381)

and should be treated this way throughout the manuscript."

The authors did not respond to this important issue through two rounds of the review process. To make myself more clear*: Prunus humilis* Bunge is the internationally accepted botanical name of the study species and *Cerasus humilis* (Bunge) S.Ya.Sokolov is a synonym. Therefore, *P. humilis* should be used through the text, while its synonym can be written in parenthesis first time it is mentioned in the Introduction section.

Moreover, the authors did not respond to the Editor's comment:

"The authors show lack of familiarity with the study species since they did not perform literature survey in order to compare the results they have obtained with the ones already published, e.g.:

Shuang, C. H. E. N. G. (2007). Free Radical Scavenging Activities of Polyphenols from *Prunus humilis* Bge Fruit [J]. Food Science, 9.

Wu, Q., Yuan, R. Y., Feng, C. Y., Li, S. S., & Wang, L. S. (2019). Analysis of polyphenols composition and antioxidant activity assessment of Chinese dwarf cherry (*Cerasus humilis* (bge.) Sok.). Natural Product Communications, 14(6), 1934578X19856509.

and other relevant articles."

I find the adequate approach to the proper taxonomic position of the studied species highly important. In addition, meticulous literature survey which would NOT exclude the most important articles in the field while comparing obtained results reported therein is essential towards increasing the manuscript quality.

We look forward to receiving your revised manuscript.

Kind regards,

Branislav T. Šiler, Ph.D.

Academic Editor

PLOS ONE

Reviewers' comments:

Reviewer's Responses to Questions

**Comments to the Author**

1. If the authors have adequately addressed your comments raised in a previous round of review and you feel that this manuscript is now acceptable for publication, you may indicate that here to bypass the “Comments to the Author” section, enter your conflict of interest statement in the “Confidential to Editor” section, and submit your "Accept" recommendation.

Reviewer #1: (No Response)

Reviewer #2: (No Response)

2. Is the manuscript technically sound, and do the data support the conclusions?

Reviewer #1: Yes

Reviewer #2: Yes

3. Has the statistical analysis been performed appropriately and rigorously? 

Reviewer #1: Yes

Reviewer #2: Yes

4. Have the authors made all data underlying the findings in their manuscript fully available?

Reviewer #1: Yes

Reviewer #2: Yes

5. Is the manuscript presented in an intelligible fashion and written in standard English?

Reviewer #1: No

Reviewer #2: No

6. Review Comments to the Author

Reviewer #1: I have previously reviewed this manuscript raising several concerns. The authors have addressed all the points raised; however, the English of the new included parts is very poor. There are too many grammatical flaws. In particular, there are errors in grammar and sentence structure and the paper sometimes is difficult to read. I have corrected some of them, but the paper needs a careful revision and a professional English editing.

Line 40: substitute “out” with “our”

Lines 46- 49: change “ It is like peach, plum and apricot, which is an ancient tree species in China with a cultivation history dated back to around 3000 years [2]. It is grown in the north of China recorded about 13 provinces, i.e., Shanxi, Hebei, Liaoning provinces, and so on [3].” with “ Like peach, plum and apricot, it is an ancient tree species in China with a cultivation history that dates back to around 3000 years [2]. Its growth in the north of China is recorded in about 13 provinces, among which Shanxi, Hebei and Liaoning [3].”

Lines 53-54: “C. humilis plants are known as ‘calcium fruit’ in China due to their high calcium content [6].”

Lines 73-74: “[17], including Cerasus humilis, represent” change with ““[17]. In this view, Cerasus humilis represents”

Line 87: “In the early research,” change with “Previously”

Line 93: change “assessions” with “accessions”

Lines 105-106: “The growth period was fertilized twice a year,” change with “During the growth period, plants were fertilized twice a year”

Lines 107-108: “From May to August watering once or twice a month, in the early November enough water was irrigated for overwintering.” Change with “From May to August the plants were watered once or twice a month; in early November the plants were irrigated with enough water for wintering.”

Lines 186: Trolox evaluation

Lines 394-397: “It is well known that fruit weight in different area, but under similar cultivation conditions in the same area also produce differernt weights, it mean that it has a close relationship between the fruit weight and accessions.” change with “It is well known that fruit weight in different areas but similar cultivation conditions produce fruits with different weights; it means that fruit weight depends on accessions and not geographic areas”.

Line 400: the better

Line 403: “our fruits are with high soluble solid content.” change with “our fruits have higher soluble solid content.”

Line 422:” a total of 31 polyphenols were detected” change with “a total of 31 polyphenols was detected” or “31 polyphenols were detected”

Line 424: ranging

Line 426: significantly and positively

Line451: About the research of sweet cherry plants,

Lines 453-454: “it is strong positive correlations between fruit phenolic content [18]” change with “that was strongly and positively correlated with fruit phenolic content [18].”

Line 457: “However, in our study” instead of “But in our study”

Lines 472-476: “So there are differences in the ability on antioxidant indices in different species. It may be that there are more components in C.humilis fruit which have the ability of scavenging ABTS free radicals, so that the C.humilis fruit has a stronger ability of scavenging ABTS free radicals.” Change with “So different species differ in antioxidant capacity. In particular, C.humilis has a stronger ability of scavenging ABTS free radicals probably because contain a higher number of different antioxidant metabolites.”. Moreover, the name of species must be italicized.

The results and discussion now appear reliable and also interesting, but the paper needs a careful revision and a professional English editing.

Reviewer #2: The article has been corrected following reviewers’ indications. Now it is acceptable for its publication after minor revision.

New paragraphs have been added to the manuscript. Please review English in these new parts of the article mainly the part that has been added to the discussion.

For example: Line 396 ..., it mean must be it means

Line 400 ...content ,the beter..... replace by: content, the better...

Line 400…quality. thus.. replace by …Thus..

Line 424 and content tanging… replace by ranging

Lines 475 and 476 C. humilis must be in italics.

These are just examples, please review in detail and correct some expressions and sentences, because it is not always easy to understand.

7. PLOS authors have the option to publish the peer review history of their article (what does this mean?). If published, this will include your full peer review and any attached files.

Reviewer #1: **Yes: **Petronia Carillo

Reviewer #2: No

---

## [Author Response · Author response to Decision Letter 3]

2 Dec 2020

Dear edit:we have upload the file incliding cover letter, manuscript, revised manuscript with track changes, and response to reviewer, and figures and supplements.

---

## [Editor Report · Decision Letter 4]

4 Dec 2020

PONE-D-20-31344R4

Fruit quality and antioxidant potential of Chinese dwarf cherry (Prunus humilis) accessions

PLOS ONE

Dear Dr. Fu,

Thank you for submitting your manuscript to PLOS ONE. After careful consideration, we feel that it has merit but does not fully meet PLOS ONE’s publication criteria as it currently stands. Therefore, we invite you to submit a revised version of the manuscript that addresses the points raised during the review process.

As I commented in the previous review round, full scientific name of the studied species must stand. Hence, "*Prunus humilis* Bunge" must stand in the main title (without quotation marks), and in the first mention in the Introduction section. There (in the Introduction section), its synonym "*Cerasus humilis *(Bunge) S.Ya.Sokolov" should be provided in parenthesis (in brackets) of the actual scientific name.

One more most important point: Figure 1, Figure 2 and Figure 6 are of very poor quality.

Labels in Fig.1 and Fig.6 are not visible. Resolution of Fig.2 is too low.

We look forward to receiving your revised manuscript.

Kind regards,

Branislav T. Šiler, Ph.D.

Academic Editor

PLOS ONE

---

## [Author Response · Author response to Decision Letter 4]

4 Dec 2020

Dear editor:

Thanks for your understand my mood about my degree, and we have revised my manuscript according your comments, and we have update and upload the figures.

Wish you have a good life!

Sincerely!

---

## [Editor Report · Decision Letter 5]

7 Dec 2020

PONE-D-20-31344R5

Fruit quality and antioxidant potential of Prunus humilis Bunge accessions

PLOS ONE

Dear Dr. Fu,

Thank you for submitting your manuscript to PLOS ONE. After careful consideration, we feel that it has merit but does not fully meet PLOS ONE’s publication criteria as it currently stands. Therefore, we invite you to submit a revised version of the manuscript that addresses the points raised during the review process.

Authors did not make visible labels in Figure 2. Since the information presented there is hardly readable, I suggest moving the Figure 2 into the Supplementary material. Figures 3 - 6 must be renamed consecutively and their references in the text should be updated too.

We look forward to receiving your revised manuscript.

Kind regards,

Branislav T. Šiler, Ph.D.

Academic Editor

PLOS ONE

---

## [Author Response · Author response to Decision Letter 5]

7 Dec 2020

Dear editor: we have moFved Figure 2 into supplement and changed figures 3-8 to 2-7.

Thanks for your comments!

---

## [Editor Report · Decision Letter 6]

9 Dec 2020

PONE-D-20-31344R6

Fruit quality and antioxidant potential of Prunus humilis Bunge accessions

PLOS ONE

Dear Dr. Fu,

Thank you for submitting your manuscript to PLOS ONE. After careful consideration, we feel that it has merit but does not fully meet PLOS ONE’s publication criteria as it currently stands. Therefore, we invite you to submit a revised version of the manuscript that addresses the points raised during the review process.

Dear authors, I owe you a big apology. In the previous review round I asked you to move Figure 2 into the supplementary material, but I really meant Figure 1. Can you please put back Figure 2 and move Figure 1 to supplementary? Once again, I deeply regret for asking you for another revision.

We look forward to receiving your revised manuscript.

Kind regards,

Branislav T. Šiler, Ph.D.

Academic Editor

PLOS ONE

---

## [Author Response · Author response to Decision Letter 6]

9 Dec 2020

Dear editor:

we have revised the figure 1 to the Fig S1, and we update the figures 1-7.

Thank you for your support!

---

## [Editor Report · Decision Letter 7]

10 Dec 2020

Fruit quality and antioxidant potential of Prunus humilis Bunge accessions

PONE-D-20-31344R7

Dear Dr. Fu,

We’re pleased to inform you that your manuscript has been judged scientifically suitable for publication and will be formally accepted for publication once it meets all outstanding technical requirements.

Kind regards,

Branislav T. Šiler, Ph.D.

Academic Editor

PLOS ONE
---

## [Editor Report · Acceptance letter]

14 Dec 2020

PONE-D-20-31344R7 

Fruit quality and antioxidant potential of *Prunus humilis* Bunge accessions 

Dear Dr. Fu:

I'm pleased to inform you that your manuscript has been deemed suitable for publication in PLOS ONE. Congratulations! Your manuscript is now with our production department. 

Kind regards, 

on behalf of

Dr. Branislav T. Šiler 

Academic Editor

PLOS ONE